# Analyzing and Optimizing Perturbation of DP-SGD Geometrically

## Abstract

This work optimizes DP-SGD from a geometric perspective. Differential privacy (DP) has become a prevalent privacy model in a wide range of machine learning tasks, especially after the debut of DP-SGD. However, DP-SGD, which directly perturbs gradients in the training iterations, fails to mitigate the negative impacts of noise on gradient direction. As a result, DP-SGD is often inefficient. In this work, we first generalize DP-SGD and theoretically derive the impact of DP noise on the training process. Our analysis reveals that, in terms of a perturbed gradient, only the noise on a direction has eminent impact on the model efficiency while that on magnitude can be mitigated by optimization techniques, i.e., fine-tuning gradient clipping and learning rate. Besides, we confirm that traditional DP introduces biased noise on the direction when adding unbiased noise to the gradient itself. Overall, the perturbation of DP-SGD is actually sub-optimal from a geometric perspective. Motivated by this, we design a geometric perturbation strategy GeoDP within the DP framework, which perturbs the direction and the magnitude of a gradient, respectively. By directly reducing the noise on the direction, GeoDP mitigates the negative impact of DP noise on model efficiency with the same DP guarantee. Extensive experiments on two public datasets (i.e., MNIST and CIFAR-10), one synthetic dataset and three prevalent models (i.e., Logistic Regression, CNN and ResNet) confirm the effectiveness and generality of our strategy.

## 1 Introduction

Although deep learning models have numerous applications in various domains, such as personal recommendation, smart manufacturing, and healthcare, the privacy leakage of training data from these models has become a growing concern. There are already mature attacks which successfully reveal the contents of private data from deep learning models (Carlini et al., 2021; Gong et al., 2021). For example, a white-box membership inference attack can infer whether a single data point belongs to the training dataset of a DenseNet with 82% test accuracy (Nasr et al., 2019). These attacks pose imminent threats to the wider adoption of deep learning in business sectors with sensitive data, especially in healthcare and fintech.

To address this concern, differential privacy (DP), which can provide quantitative amount of privacy preservation to individuals in the training dataset, is embraced by the most prevalent model training technique, i.e., stochastic gradient descent (SGD). Referred to as DP-SGD (Li et al., 2021; Zeighami et al., 2022; Liu et al., 2021; Bao et al., 2022), this algorithm adds random DP noise to gradients in the training process so that attackers cannot infer private data from model parameters with a high probability.

However, a primary drawback of DP-SGD is the ineffective training process caused by the overwhelming noise, which deteriorates the model efficiency. Although much attention (Abadi et al., 2016; Mironov, 2017; Fu et al., 2023) has been paid to reducing the noise scale, the majority of existing solutions, which numerically add DP noise to gradients, **do not fully exploit the geometric nature of SGD** (i.e., descending gradient to locate the optima). As reviewed in Section E.3, SGD exhibits a distinctive geometric property — the direction of a gradient rather than the magnitude determines the descent trend. By contrast, regular DP algorithms, such as the Gaussian mechanism (Dwork et al., 2014), were originally designed to preserve numerical (scalar) values instead

| Symbol | Meaning |
|--------|---------|
| $\epsilon$ | privacy budget |
| $\beta$ | bounding factor |
| $\sigma$ | noise multiplier |
| $\boldsymbol{w}^\star$ | global optima |
| $\tilde{\boldsymbol{g}}$ | clipped gradient |
| $\boldsymbol{n}$ | DP noise vector |
| $\boldsymbol{g}^*$ | perturbed gradient from traditional DP |
| $\boldsymbol{g}^\star$ | perturbed gradient from GeoDP |
| $\boldsymbol{\theta}$ | direction of a gradient |
| $\|\boldsymbol{g}\|$ | magnitude of a gradient |

Table 1: Frequently-used notations

of vector values, which causes at least two limitations in DP-SGD. First, **existing optimization techniques of SGD (i.e., fine-tuning clipping and learning rate)**, which can effectively reduce the noise on the magnitude of a gradient, **cannot alleviate the negative impact on the direction**, as illustrated by Example 1. Second, **traditional DP introduces biased noise on the direction of a gradient**, even if the total noise to the gradient is unbiased (proved in Lemma 1 of this paper). As a result, the perturbation of traditional DP-SGD is only sub-optimal from a geometric perspective.

**Example 1** *Suppose that we have a two-dimensional gradient $\boldsymbol{g} = (1, \sqrt{3})$ with its direction $\boldsymbol{\theta} = \arctan(\sqrt{3}/1) = \pi/3$ and magnitude $\|\boldsymbol{g}\| = \sqrt{1+3} = 2$. Given clipping threshold $C_1 = 2$, we add noise $\boldsymbol{n}_1 = (0.3, 0.15)$ to the clipped gradient $\tilde{\boldsymbol{g}}_1 = \boldsymbol{g}/\max\{1, \|\boldsymbol{g}\|/C_1\} = (1, \sqrt{3})$ and derive the perturbed direction $\boldsymbol{\theta}_1^* = \arctan\frac{\sqrt{3}+0.15}{1+0.3} \approx 0.97$. If $C_2 = 1$, the clipped gradient and the noise would be $\tilde{\boldsymbol{g}}_2 = \boldsymbol{g}/\max\{1, \|\boldsymbol{g}\|/C_2\} = (\frac{1}{2}, \frac{\sqrt{3}}{2})$ and $\boldsymbol{n}_2 = \boldsymbol{n}_1/(C_1/C_2) = (0.15, 0.075)$, respectively, as per DP-SGD (Abadi et al., 2016). Still, the perturbed direction is $\boldsymbol{\theta}_2^* = \arctan\frac{\frac{\sqrt{3}}{2}+0.075}{\frac{1}{2}+0.15} \approx 0.97$. Although the noise scale is successfully reduced by gradient clipping ($\|\boldsymbol{n}_2\| < \|\boldsymbol{n}_1\|$), the perturbation on the direction of a gradient remains the same ($\boldsymbol{\theta}_2^* = \boldsymbol{\theta}_1^*$).*

In this paper, we propose a geometric perturbation strategy GeoDP to address these limitations. First, we theoretically derive the impact of DP noise on the efficiency of DP-SGD. Based on this fine-grained analysis, the perturbation of DP-SGD, which introduces biased noise to the direction of a gradient, is actually sub-optimal. Inspired by this, we propose a geometric perturbation strategy *GeoDP* which separately perturbs the direction and the magnitude of a gradient, so as to mitigate the noisy gradient direction and optimize model efficiency with the same DP guarantee. In summary, our main contributions are as follows:

- To the best of our knowledge, we are the first to prove that the perturbation of traditional DP-SGD is actually sub-optimal from a geometric perspective.

- Within the classic DP framework, we propose a geometric perturbation strategy *GeoDP* to directly add the noise on the direction of a gradient, which rigorously guarantees a better trade-off between privacy and efficiency.

- Extensive experiments on public datasets as well as prevalent AI models validate the generality and effectiveness of GeoDP.

The rest of this paper is organized as follows. Section 2 introduces basic concepts as well as problem formulation. Section 3 presents our theoretical analysis on the deficiency of DP-SGD while Section 4 presents the perturbation strategy *GeoDP*. Experimental results are in Section 5, followed by a conclusion in Section 7.

## 2 PRELIMINARIES AND PROBLEM FORMULATION

### 2.1 DIFFERENTIAL PRIVACY

Differential Privacy (DP) is a mathematical framework that quantifies the privacy preservation. Formally, $(\epsilon, \delta)$-*DP* is defined as follows:

**Definition 1** (($\epsilon,\delta$)-DP). *A randomized algorithm $\mathcal{M} : D \to R$ satisfies ($\epsilon,\delta$)-DP if for all datasets $D$ and $D'$ differing on a single element, and for all subsets $S$ of $R$, the following inequality always holds:*

$$\Pr[\mathcal{M}(D) \in S] \le e^{\epsilon} \times \Pr[\mathcal{M}(D') \in S] + \delta. \tag{1}$$

In essence, DP guarantees that given any outcome of $\mathcal{M}$, it is unlikely for any third party to infer the original record with high confidence. Privacy budget $\epsilon$ controls the level of preservation. Namely, a lower $\epsilon$ means stricter privacy preservation and thus poorer efficiency, and vice versa. $\delta$ determines the probability of not satisfying $\epsilon$ preservation.

To determine the noise scale of DP, we measure the maximum change of $\mathcal{M}$ for $L_2$-norm as:

**Definition 2** ($L_2$-sensitivity). *The $L_2$-sensitivity of $\mathcal{M}$ is:*

$$\Delta\mathcal{M} = \max_{\|D-D'\|_1=1} \|\mathcal{M}(D) - \mathcal{M}(D')\|_2. \tag{2}$$

Through out the paper, we follow the common practice of existing works (Abadi et al., 2016; Fu et al., 2023) and use Gaussian mechanism (Dwork et al., 2014) for theoretical analysis and experiments. The perturbed value of Gaussian mechanism is $g^* = g + Gau(0, 2\Delta\mathcal{M} \ln \frac{1.25}{\delta}/\epsilon^2)$, where $Gau$ denotes a random variable that follows Gaussian distribution with probability density function $f(x) = \frac{1}{\sigma\sqrt{2\pi}} \exp(-\frac{(x-\mu)^2}{2\sigma^2})$. Referring to the standard deviation of $Gau(0, 2 \ln \frac{1.25}{\delta}/\epsilon^2)$ as **the noise multiplier $\sigma$**, **the noise scale** of Gaussian mechanism is $\Delta\mathcal{M}\sigma$ (Dwork et al., 2014).

### 2.2 STOCHASTIC GRADIENT DESCENT

SGD (stochastic gradient descent) is one of the most widely used optimization techniques in machine learning (Bottou, 2012). Let $D$ be the private dataset, and $\boldsymbol{w}$ denote the model parameters (a.k.a the training model). Given $S \subseteq D$ and $S = \{s_1, s_2, ..., s_{(B-1)}, s_B\}$ ($B$ denoting the number of data in $S$), the objective $F(\boldsymbol{w})$ can be formulated as $F(\boldsymbol{w}; S) = \frac{1}{B} \sum_{j=1}^{B} l(\boldsymbol{w}; s_j)$, where $l(\boldsymbol{w}; s_j)$ is the loss function trained on one subset data $s_j$ to optimize $\boldsymbol{w}$.

To optimize this task, we follow the common practice of existing works and use mini-batch SGD (LeCun et al., 2002). Given the total number of iterations $T$, $\boldsymbol{w}_t = (\boldsymbol{w}_{t1}, \boldsymbol{w}_{t2}, ..., \boldsymbol{w}_{t(d-1)}, \boldsymbol{w}_{td})$ ($0 \le t \le T-1$) denotes a $d$-dimensional model weight derived from the $t$-th iteration (where $t = 0$ is the initiate state). While using $\eta$ to denote the learning rate, we have the gradient $\boldsymbol{g}_t$ of the $t$-th iteration:

$$\boldsymbol{g}_t = \nabla F(\boldsymbol{w}_t; S) = \frac{1}{B} \sum_{j=1}^{B} \nabla l(\boldsymbol{w}; s_j) = \frac{1}{B} \sum_{j=1}^{B} \boldsymbol{g}_{tj}. \tag{3}$$

where $\nabla l = \left( \frac{\partial l}{\partial \boldsymbol{w}_1}, \frac{\partial l}{\partial \boldsymbol{w}_2}, ..., \frac{\partial l}{\partial \boldsymbol{w}_{d-1}}, \frac{\partial l}{\partial \boldsymbol{w}_d} \right)$, and respective gradients $\{\boldsymbol{g}_{tj} | 1 \le j \le B\}$ are derived from respective data $\{s_j | 1 \le j \le B\}$ of the batch. The $t$-th iteration updates the model weight $\boldsymbol{w}_{t+1}$ as $\boldsymbol{w}_{t+1} = \boldsymbol{w}_t - \eta\boldsymbol{g}_t$.

By tuning the batch size $B$, the analysis on this optimization technique also applies to its variants. For example, if $B = |D|$, it is equivalent to the batch gradient descent (Boyd & Vandenberghe, 2004); if $B = 1$, it is equivalent to the stochastic gradient descent (Bottou, 2012). Throughout this paper, we abbreviate mini-batch stochastic gradient descent and its variants collectively as SGD.

SGD is known to have an intrinsic problem of gradient explosion (Pascanu et al., 2013). It often occurs when the gradients become very large during backpropagation, and causes the model to converge rather slowly. As the most effective solution to this problem, gradient clipping (Pascanu et al., 2013) is also considered in this work. Let $\|\boldsymbol{g}\|$ denote the $L_2$-norm of a $d$-dimensional vector $\boldsymbol{g} =$

$(g_1, g_2, ..., g_{d-1}, g_d)$, i.e., $\|g\| = \sqrt{\sum_{z=1}^d g_z^2}$. Assume that $G$ is the maximum $L_2$-norm value of all possible gradients for any weight $w$ derived from any subset $S$, i.e., $G = \sup_{w \in \mathbb{R}^d, S \in D} \mathbb{E}[\|g\|]$. Then each gradient $g$ is clipped by a clipping threshold $C \in (0, G]$. Formally, the clipped gradient $\tilde{g}$ is:

$$\tilde{g} = \frac{g}{\max\{1, \|g\|/C\}}. \tag{4}$$

Another advantage of clipping is to reduce the sensitivity of a gradient, which therefore decreases the noise addition in DP-SGD. The most recent state-of-the-art work proposes AUTO-S (Bu et al., 2024) for automatic clipping, which conducts clipping as follows $\tilde{g} = \frac{g}{\|g\|+0.01}$.

Applying Equation 4 to Equation 3, we derive the clipped gradient from the $t$-th iteration as $\tilde{g}_t = \frac{1}{B} \sum_{j=1}^B \tilde{g}_{tj}$.

### 2.3 Problem Formulation of DP-SGD

As shown in Algorithm 2 of Appendix A , in each iteration of DP-SGD, $w_{t+1}$ is perturbed to $w_{t+1}^*$ by adding DP noise $n_t$ to the sum of $\tilde{g}_{tj}$. Let $g_t^*$ denote the perturbed gradient. Formally, we have $g_t^* = \frac{1}{B}(\sum_{j=1}^B \tilde{g}_{tj} + n_t) = \tilde{g}_t + n_t/B$ and $w_{t+1}^* = w_t - \eta g_t^*$. Accordingly, the following definition establishes the measurement for model efficiency (ME). Obviously, a smaller ME means a better model efficiency.

**Definition 3** (*Model Efficiency (ME)*). *Suppose there exists a global optima $w^\star$, the model deficiency can be measured by the Euclidean Distance between the current model $w_{t+1}^*$ and the optima $w^\star$, i.e., Model efficiency (ME) = $\left\|w_{t+1}^* - w^\star\right\|^2$.*

As having to validate the optimality of GeoDP over DP on preserving the descent trend, we follow the common practice (Wang et al., 2019) and adopt mean square error (MSE) to measure the error on perturbed directions. In general, a larger MSE means a larger perturbation.

**Definition 4** (*Mean Square Error (MSE)*). *Considering the perturbed directions $\{\theta_1^*, \theta_2^*, ..., \theta_{m-1}^*, \theta_m^*\}$ and the original directions $\{\theta_1, \theta_2, ..., \theta_{m-1}, \theta_m\}$ of $m$ gradients, MSE of perturbed directions is defined as $MSE(\theta^*) = \frac{1}{m} \sum_{i=1}^m \|\theta_i^* - \theta_i\|_2^2$.*

The problem in this work is to investigate the impact of DP noise $n_t$ on the SGD efficiency, i.e., $\left\|w_{t+1}^* - w^\star\right\|^2$, and further optimize the model efficiency by reducing the noise on the direction of a gradient, i.e., reducing $MSE(\theta^*)$.

## 3 Deficiency of DP-SGD: a gap between directional SGD and numerical DP

In this section, we identify an intrinsic deficiency in DP-SGD. Let the trained models of DP-SGD and non-private SGD be denoted by $w_{t+1}^* = w_t - \eta \tilde{g}_t^*$ and $w_{t+1} = w_t - \eta \tilde{g}_t$, respectively. The Euclidean distances between the current models and the global optima (i.e., $\left\|w_{t+1}^* - w^\star\right\|^2$ and $\left\|w_{t+1} - w^\star\right\|^2$) reflect the model efficiency of DP-SGD and non-private SGD, respectively. Apparently, the smaller this distance is, the better efficiency the model achieves. Their efficiency difference (ED) (i.e., $\left\|w_{t+1}^* - w^\star\right\|^2 - \left\|w_{t+1} - w^\star\right\|^2$), on the other hand, can describe the impact of DP noise on the model efficiency, as presented by the following theorem.

**Theorem 1** (*Impact of DP Noise on Model Efficiency*). *Suppose $n_\sigma$ follows a noise distribution with the standard deviation $\sigma I$, ED can be measured as $\left\|w_{t+1}^* - w^\star\right\|^2 - \left\|w_{t+1} - w^\star\right\|^2$*

$$= \eta^2 \underbrace{\left(\frac{2C}{B}\langle n_\sigma, \tilde{g}_t\rangle + \frac{C^2 n_\sigma^2}{B^2}\right)}_{Item \quad A} + \frac{2\eta C}{B} \underbrace{\langle n_\sigma, w^\star - w_t\rangle}_{Item \quad B}.$$

**Proof 1** *See Appendix C.1 for details.*

In general, we wish the efficiency of DP-SGD closer to SGD, i.e., to make ED as close to zero as possible. This theorem coincides with many empirical findings in existing works. Item A, for example, shows that the introduction of DP noise would cause a bias to the global optima. That is, **DP-SGD cannot stably converge to the global optima, while sometimes reaching that point**, as proved by Corollary 1. This means that the model efficiency of DP-SGD is always lower than regular SGD (Xia et al., 2023; Zhang et al., 2022; Chen et al., 2020; Tang et al., 2024). In practice, in order to provide a better model efficiency, existing works (Abadi et al., 2016; Yu et al., 2019; Feng et al., 2020) apply lower noise scale (i.e., smaller $n_\sigma$) when DP-SGD is about to converge. This operation makes Item A close to zero (but normally non-zero). Another example is that large batch size can enhance the efficiency of DP-SGD, as it can certainly reduce both Item A and Item B (Fu et al., 2023).

**Corollary 1** *DP-SGD cannot stably stay at global optima.*

**Proof 2** *See Appendix C.2 for details.*

More importantly, this theorem reveals that DP-SGD techniques, such as adaptive clipping and learning rate, are incapable of counteracting the impact of DP noise on the direction of a gradient. On one hand, **Item A describes how the noise scale impacts the model efficiency**. To reduce this impact, small learning rate ($\eta^2$) and clipping threshold ($C$ and $C^2$), or large batch size $B$ is effective. This conclusion is confirmed by many existing works, as reviewed in Section 6. On the other hand, **Item B**, the inner product between the noise $n_t$ and the training process ($w^\star - w_t$ can be considered as the distance for SGD to descend, i.e., descent trend) **reflects how the perturbation impacts the further training**. While capable of reducing Item A, fine-tuning hyper-parameters cannot reduce Item B, as proved by the following corollary.

**Corollary 2** *optimization techniques of DP-SGD (i.e., fine-tuning clipping and learning rate) cannot reduce the impact of noise on the gradient direction.*

**Proof 3** *See Appendix C.3 for details.*

In general, this corollary points out an intrinsic deficiency of DP-SGD. That is, as a gradient is actually a vector instead of a numerical array, **traditional DP mechanisms**, which add noise to values of a gradient, **cannot directly reduce the noise on gradient direction (Item B)**. Even worse, **DP introduces biased noise to the direction, while adding unbiased noise to the gradient itself,** as further proved via hyper-spherical coordinate system (see Lemma 1 for rigorous proofs).

# 4 GEOMETRIC PERTURBATION: GEODP

In the previous analysis, we have proved the sub-optimality of traditional DP-SGD. In this section, we seize this opportunity to **perturb the direction and the magnitude of a gradient, respectively, so that the noise on gradient direction is directly reduced**. Within the DP framework, our strategy significantly improves the model efficiency.

In what follows, we first introduce $d$-spherical coordinate system (Thomas & Weir, 2006) in Section 4.1, where one $d$-dimensional gradient is converted to one magnitude and one direction. By perturbing gradients in the $d$-spherical coordinate system, we propose our perturbation strategy *GeoDP* to optimize the model efficiency in Section 4.2. Privacy and efficiency analysis is provided to prove its compliance with DP definition and huge advantages over DP-SGD in Section 4.3.

## 4.1 HYPER-SPHERICAL COORDINATE SYSTEM

The $d$-spherical coordinate system (Thomas & Weir, 2006), also known as the hyper-spherical coordinate system, is commonly used to analyze geometric objects in high-dimensional space, e.g., the gradient. Compared to the rectangular coordinate system (Thomas & Weir, 2006), such a system directly represents any $d$-dimensional vector $g = (g_1, g_2, ..., g_{d-1}, g_d)$ using a magnitude $\|g\|$ and a direction $\theta = (\theta_1, \theta_2, ..., \theta_{d-2}, \theta_{d-1})$. Formally, the magnitude is:

$$\|g\| = \sqrt{\sum_{z=1}^{d} g_z^2}. \tag{5}$$

and its direction $\boldsymbol{\theta}$ is:

$$\boldsymbol{\theta}_z = \begin{cases} \arctan2\left(\sqrt{\sum_z^{d-1} \boldsymbol{g}_{z+1}^2}, \boldsymbol{g}_z\right) & \text{if } 1 \le z \le d-2, \\ \arctan2\left(\boldsymbol{g}_{z+1}, \boldsymbol{g}_z\right) & \text{if } z = d-1. \end{cases} \tag{6}$$

where arctan2 is the two-argument arctangent function defined as follows:

$$\arctan2(y, x) = \begin{cases} \arctan\left(\frac{y}{x}\right) & \text{if } x > 0, \\ \arctan\left(\frac{y}{x}\right) + \pi & \text{if } x < 0 \text{ and } y \ge 0, \\ \arctan\left(\frac{y}{x}\right) - \pi & \text{if } x < 0 \text{ and } y < 0, \\ \frac{\pi}{2} & \text{if } x = 0 \text{ and } y > 0, \\ -\frac{\pi}{2} & \text{if } x = 0 \text{ and } y < 0, \\ \text{undefined} & \text{if } x = 0 \text{ and } y = 0. \end{cases} \tag{7}$$

While having the same functionality as arctan, arctan2 is more robust. For example, arctan2 can deal with a zero denominator ($\boldsymbol{g}_z = 0$). Note that $\sqrt{\sum_z^{d-1} \boldsymbol{g}_{z+1}^2}$ in Equation 6 is always non-negative. For $1 \le z \le d-2$, the range of $\arctan2\left(\sqrt{\sum_z^{d-1} \boldsymbol{g}_{z+1}^2}, \boldsymbol{g}_z\right)$ is either $\left(0, \frac{\pi}{2}\right]$ or $\left(\frac{\pi}{2}, \pi\right)$ if $\boldsymbol{g}_z \ge 0$ or $\boldsymbol{g}_z < 0$, as per Equation 7. **As such, the range of $\boldsymbol{\theta}_{1 \le z \le d-2}$ is $(0, \pi)$. For $z = d-1$, the range of $\boldsymbol{\theta}_z$ is $(-\pi, \pi)$ as per Equation 7.**

We can also convert a vector $(\|\boldsymbol{g}\|, \boldsymbol{\theta})$ in $d$-spherical coordinates back to rectangular coordinates $(\boldsymbol{g}_1, \boldsymbol{g}_2, ..., \boldsymbol{g}_{d-1}, \boldsymbol{g}_d)$ using the following equation:

$$\boldsymbol{g}_z = \begin{cases} \|\boldsymbol{g}\| \cos \boldsymbol{\theta}_z, & \text{if } z = 1 \\ \|\boldsymbol{g}\| \prod_{i=1}^{z-1} \sin \boldsymbol{\theta}_i \cos \boldsymbol{\theta}_z, & \text{if } 2 \le z \le d-1 \\ \|\boldsymbol{g}\| \prod_{i=1}^{z-1} \sin \boldsymbol{\theta}_i, & \text{if } z = d \end{cases} \tag{8}$$

Figure 2 in Appendix B provides an example of conversions in three-dimensional space. Given $\|\boldsymbol{g}\| = \sqrt{\boldsymbol{g}_1^2 + \boldsymbol{g}_2^2 + \boldsymbol{g}_3^2}$, $\boldsymbol{\theta}_1 = \arctan2\left(\sqrt{\boldsymbol{g}_2^2 + \boldsymbol{g}_3^2}, \boldsymbol{g}_1\right)$ and $\boldsymbol{\theta}_2 = \arctan2\left(\boldsymbol{g}_3, \boldsymbol{g}_2\right)$, a vector $\boldsymbol{g} = (\boldsymbol{g}_1, \boldsymbol{g}_2, \boldsymbol{g}_3)$ in rectangular coordinate system (marked in black) can be represented as $(\|\boldsymbol{g}\|, \boldsymbol{\theta}_1, \boldsymbol{\theta}_2)$ in hyper-spherical coordinate system (marked in blue). Without loss of generality, we use $\boldsymbol{g} \leftrightarrow (\|\boldsymbol{g}\|, \boldsymbol{\theta})$ to denote the reversible conversions between two systems.

### 4.2 GEODP—GEOMETRIC DP PERTURBATION FOR DP-SGD

*GeoDP* directly reduces the noise on the gradient direction via $d$-spherical coordinate system. Algorithm 1 describes how *GeoDP* works, and major steps are interpreted as follows:

1. *Spherical-coordinate Conversion:* Convert the clipped gradient to hyper-spherical coordinate system according to Equation 5 and Equation 6, i.e., $\boldsymbol{g} \to (\|\boldsymbol{g}\|, \boldsymbol{\theta})$, which allows perturbation on the magnitude and the direction of a gradient, respectively.

2. *Reducing the Direction Range (Sensitivity):* According to Theorem .2, the averaged direction of gradients $\{\tilde{\boldsymbol{g}}_{tj} | 1 \le j \le B\}$ should be centered at one small range, rather than uniformly spreading the whole vector space. This conclusion is also confirmed by various SGD studies (Yu et al., 2019; Bottou, 2012). DP-SGD, taking the whole direction space as the privacy region, is therefore overprotective and low efficient. In this work, a bounding factor $\beta \in (0, 1]$ defines the privacy region into a subspace around the original direction, which significantly reduces the noise addition in Step 3. For $1 \le z < d-1$, given $0 \le \Gamma_1 \le \boldsymbol{\theta}_z \le \Gamma_2 \le \pi$, $\beta$ determines the range between $\Gamma_1$ and $\Gamma_2$, i.e., $\Gamma_2 - \Gamma_1 = \Delta\boldsymbol{\theta}_z = \beta\pi$. Similarly, $\Gamma_2 - \Gamma_1 = \Delta\boldsymbol{\theta}_z = 2\beta\pi$ for $z = d-1$. Note that $\beta = 1$ means the full space. This parameter directly determines the sensitivity of the direction, which consequently influences the noise addition in the following step.

3. *Noise Addition:* GeoDP allows to perturb the magnitude and the direction of a gradient, respectively. For the magnitude, $\|\tilde{\boldsymbol{g}}_t\|$ is already bounded by $C$ in the first stage. Similar to DP-SGD, the noise scale of the perturbed magnitude is $C\sigma$. For the direction, the noise

scale is the sensitivity $\Delta\boldsymbol{\theta}$ times the noise multiplier $\sigma$. Note that maximum changes of $\tilde{\boldsymbol{\theta}}_{1 \leq z \leq d-2}$ and $\tilde{\boldsymbol{\theta}}_{d-1}$ are $\beta\pi$ and $2\beta\pi$, respectively, due to the bounding of the direction range. Overall, $\Delta\boldsymbol{\theta} = \sqrt{(d-2)(\beta\pi)^2 + (2\beta\pi)^2} = \sqrt{d+2}\beta\pi$.

4. *Rectangular-coordinate Conversion:* Convert the perturbed magnitude and direction back to rectangular coordinates according to Equation 8, i.e., $\left(\|\tilde{\boldsymbol{g}}_t\|^{\star}, \boldsymbol{\theta}_t^{\star}\right) \to \tilde{\boldsymbol{g}}_t^{\star}$, which allows future gradient descent.

---

**Algorithm 1** GeoDP-SGD

---

**Require:** Batch size $B$, noise multiplier $\sigma$, clipping threshold $C$, bounding factor $\beta(0 < \beta \leq 1)$, learning rate $\eta$, total number of iterations $T$.
**Ensure:** Trained model $\boldsymbol{w}_T^{\star}$.
1: Initialize a model with parameters $\boldsymbol{w}_0$.
2: **for** each iteration $t = 0, 1, ..., T-2, T-1$ **do**
3:      Derive the average clipped gradient $\tilde{\boldsymbol{g}}_t$ with respect to the batch size $B$ and the clipping threshold $C$.
4:      Convert $\tilde{\boldsymbol{g}}_t$ to $d$-spherical coordinates as $(\|\tilde{\boldsymbol{g}}_t\|, \boldsymbol{\theta}_t)$.
5:      Bound the privacy region $\Delta$ of $\boldsymbol{\theta}$ as follows:

$$\Delta\boldsymbol{\theta}_z = \begin{cases} \Delta\boldsymbol{\theta}_{1 \leq z \leq d-2} & = \beta\pi, \\ \Delta\boldsymbol{\theta}_{d-1} & = 2\beta\pi. \end{cases}$$

6:      $\|\tilde{\boldsymbol{g}}_t\|^{\star} = \|\tilde{\boldsymbol{g}}_t\| + \frac{C}{B}\boldsymbol{n}_\sigma$, $\tilde{\boldsymbol{\theta}}_t^{\star} = \tilde{\boldsymbol{\theta}}_t + \frac{\sqrt{d+2}\beta\pi}{B}\boldsymbol{n}_\sigma$, where $\boldsymbol{n}_\sigma$ follows a zero-mean Gaussian distribution with standard deviation $\sigma$.
7:      Convert $\left(\|\tilde{\boldsymbol{g}}_t\|^{\star}, \tilde{\boldsymbol{\theta}}_t^{\star}\right)$ back to rectangular coordinates as the perturbed gradient $\tilde{\boldsymbol{g}}_t^{\star}$.
8:      Update $\boldsymbol{w}_{t+1}^{\star}$ by taking a step in the direction of the noisy gradient, i.e., $\boldsymbol{w}_{t+1}^{\star} = \boldsymbol{w}_t - \eta\tilde{\boldsymbol{g}}_t^{\star}$.
9: **end for**

---

In general, GeoDP provides better efficiency to SGD from two perspectives. First, **GeoDP adds unbiased noise, whereas traditional DP introduces biased perturbation, to the gradient direction** (see Lemma 1 for rigorous proofs). This counter-intuitive conclusion is supported by the fact that traditional DP, which adds unbiased noise to the gradient itself, however accumulates noise on different angles of one direction. Example 2 demonstrates how this noise accumulation happens. As such, numerical perturbation of DP seriously degrades the accuracy of directional information. GeoDP, on the other hand, independently controls the noise on each angle and therefore prevents noise accumulation.

**Example 2** *Suppose that we have a three-dimensional gradient $\boldsymbol{g} = (\boldsymbol{g}_1, \boldsymbol{g}_2, \boldsymbol{g}_3)$. Following traditional DP, these three should be added noise $\boldsymbol{n} = (\boldsymbol{n}_1, \boldsymbol{n}_2, \boldsymbol{n}_3)$. The first angle $\boldsymbol{\theta}_1^*$ of perturbed gradient direction $\boldsymbol{\theta}^*$ should be $\arctan2\left(\sqrt{(\boldsymbol{g}_2 + \boldsymbol{n}_2)^2 + (\boldsymbol{g}_3 + \boldsymbol{n}_3)^2}, \boldsymbol{g}_1 + \boldsymbol{n}_1\right)$, according to Equation 4. It is very obvious that noise of three dimensions $(\boldsymbol{n}_1, \boldsymbol{n}_2, \boldsymbol{n}_3)$ is accumulated to the first angle $\boldsymbol{\theta}_1$, and this accumulation is biased.*

Second, via coordinates conversion, $d$-dimensional gradient is transferred to one magnitude and $d-1$ directions. By composition theory, $\frac{d-1}{d}$ privacy budget is allocated to the direction by GeoDP, which can better preserves directional information.

Finally, we discuss the time complexity of GeoDP-SGD. For DP-SGD, given the size of private dataset $|D|$ and the number of gradient's dimensions $d$, DP-SGD takes $O(|D|d)$ time to calculate derivatives in one epoch (Yu et al., 2019). By contrast, coordinate conversion costs a little time because it only involves simple geometry calculation. Besides that, GeoDP has the same time complexity as DP-SGD.

### 4.3 EFFICIENCY COMPARISON BETWEEN GEODP AND TRADITIONAL DP

Via hyper-spherical coordinate system, we can identify deficiencies of traditional DP from a geometric perspective and further understand the merits of GeoDP. If clipping threshold is fixed, the max magnitude of a clipped gradient is determined, because $\|\tilde{\boldsymbol{g}}\| = \frac{\|\tilde{\boldsymbol{g}}\|}{\max\{1, \|\boldsymbol{g}\|/C\}} \leq C$. That is,

the clipped gradients are within the hyper-sphere whose radius $R$ is $C$. For example, $\boldsymbol{g}$ (highlighted in black) is a vector within the hyper-sphere whose radius is $\|\boldsymbol{g}\|$ (highlighted in blue). By adding noise, traditional DP makes sure that any two gradients within the hyper-sphere are indistinguishable. However, there are two serious disadvantages.

**On one hand, numerical noise addition does not respect the geometric property of gradients**, as interpreted by the following example. In general, traditional DP seriously sabotages the geometric property of a gradient, which eventually results in low model efficiency.

**Example 3** *Suppose two gradients $\tilde{\boldsymbol{g}}_1 = (1, 1)$, $\tilde{\boldsymbol{g}}_2 = (2, 2)$ and clipping threshold $C = 2\sqrt{2}$. As such, these two gradients are both within $R = C = 2\sqrt{2}$ hyper-sphere, and their directions are both $\boldsymbol{\theta} = \arctan2(1, 1) = \arctan2(2, 2) = \frac{\pi}{4}$. In other words, DP adds the same scale of noise to both gradients. Assuming that DP noise $\boldsymbol{n} = (2, -1)$ is added to both gradients, the directions of both perturbed gradients are $\boldsymbol{\theta}_1^* = \arctan2(1 - 1, 1 + 2) = 0$ and $\boldsymbol{\theta}_2^* = \arctan2(2 - 1, 2 + 2) \approx \frac{2\pi}{25}$, respectively. Although directions of original gradients are the same ($\boldsymbol{\theta} = \frac{\pi}{4}$), directions of perturbed gradients ($\boldsymbol{\theta}_1^* \neq \boldsymbol{\theta}_2^* \neq \boldsymbol{\theta}$) are obviously different, even if their added noise $\boldsymbol{n} = (2, -1)$ is the same.*

**On the other hand, traditional DP, which preserves all directions equally within the hyper-sphere, adds excessive noise to the gradient.** Different from regular SGD, DP-SGD usually requires very large batch size (e.g., 16,384) to reduce the negative impact of noise (Fu et al., 2023), which makes training process less "stochastic" (Yu et al., 2019; Bottou, 2012). In particular, the summation of gradients $\{\tilde{\boldsymbol{g}}_{jz} | 1 \leq j \leq B, 1 \leq z \leq d\}$ follows *Lindeberg–Lévy Central Limit Theorem* (CLT) (Shanthikumar & Sumita, 1984) as these gradients are independently and identically distributed (each of them is derived from the same dataset). As such, we can use Gaussian distribution to model the average of this summation (i.e., $\tilde{\boldsymbol{g}}_z = \frac{1}{B} \sum_{j=1}^B \tilde{\boldsymbol{g}}_{jz}$), as proved by the following theorem.

**Theorem 2** *(Modeling of the Averaged Stochastic Gradients). Suppose that $var(\tilde{\boldsymbol{g}}_{jz})$ and $\mathbb{E}(\tilde{\boldsymbol{g}}_{jz})$ are the variance and the expectation of $\{\tilde{\boldsymbol{g}}_{jz} | 1 \leq j \leq B, 1 \leq z \leq d\}$, the asymptotic probability density function (pdf) of $\tilde{\boldsymbol{g}}_z$ over $B$ is $\lim_{B \to \infty} f(\tilde{\boldsymbol{g}}_z) = \sqrt{\frac{B}{2\pi * var(\tilde{\boldsymbol{g}}_{jz})}} \exp\left(-\frac{B^2 * (x - \mathbb{E}(\tilde{\boldsymbol{g}}_{jz}))^2}{2 * var(\tilde{\boldsymbol{g}}_{jz})}\right)$.*

**Proof 4** *See Appendix C.4 for details.*

As suggested by Theorem .2, large batch size would incur unevenly distributed average of gradients, making the training process less stochastic. It also proves that the average stochastic gradients concentrates at a certain direction, rather than evenly spreading in the whole vector space. As such, traditional DP-SGD, only effective in the whole vector space, wastes privacy budgets by preserving unnecessary directions. In contrast, GeoDP preserves the subspace where directions of various gradients are concentrated, and therefore provides better efficiency, as jointly proved by the following lemma (which indicates the better accuracy of GeoDP on preserving directional information) and theorem (which further indicates the superiority of GeoDP on model efficiency). Experimental results in Section D.2 also confirm our analysis.

**Lemma 1** *Given the original direction $\boldsymbol{\theta}$, two perturbed directions $\boldsymbol{\theta}^\star$ and $\boldsymbol{\theta}^*$ from GeoDP and DP, respectively, there always exists such a bounding factor $\beta$ that $MSE(\hat{\boldsymbol{\theta}}_t^\star) < MSE(\hat{\boldsymbol{\theta}}_t^*)$ holds.*

**Proof 5** *See Appendix C.5 for details.*

We further prove the optimality of GeoDP to traditional DP in the efficiency of SGD tasks in the following theorem.

**Theorem 3** *(Optimality of GeoDP). Let $\boldsymbol{w}_{t+1}^\star = \boldsymbol{w}_t - \eta \tilde{\boldsymbol{g}}_t^\star$, $\boldsymbol{w}_{t+1}^* = \boldsymbol{w}_t - \eta \tilde{\boldsymbol{g}}_t^*$ and $\tilde{\boldsymbol{g}}_t$, $\tilde{\boldsymbol{g}}_t^\star$ and $\tilde{\boldsymbol{g}}_t^*$ be the clipped gradient, noisy gradients of GeoDP and DP, respectively. Besides, $\tilde{\boldsymbol{g}}_t \to \left(\|\tilde{\boldsymbol{g}}_t\|, \tilde{\boldsymbol{\theta}}_t\right)$, $\tilde{\boldsymbol{g}}_t^\star \to \left(\|\tilde{\boldsymbol{g}}_t\|^\star, \tilde{\boldsymbol{\theta}}_t^\star\right)$ and $\tilde{\boldsymbol{g}}_t^* \to \left(\|\tilde{\boldsymbol{g}}_t\|^*, \tilde{\boldsymbol{\theta}}_t^*\right)$. The following inequality always holds if $\tilde{\boldsymbol{g}}_t^\star$ and $\tilde{\boldsymbol{g}}_t^*$ both follow $(\epsilon, \delta)$-DP, i.e., $\mathbb{E}\left(\|\boldsymbol{w}_{t+1}^\star - \boldsymbol{w}^\star\|^2\right) < \mathbb{E}\left(\|\boldsymbol{w}_{t+1}^* - \boldsymbol{w}^\star\|^2\right)$.*

**Proof 6** *See Appendix C.6 for details.*

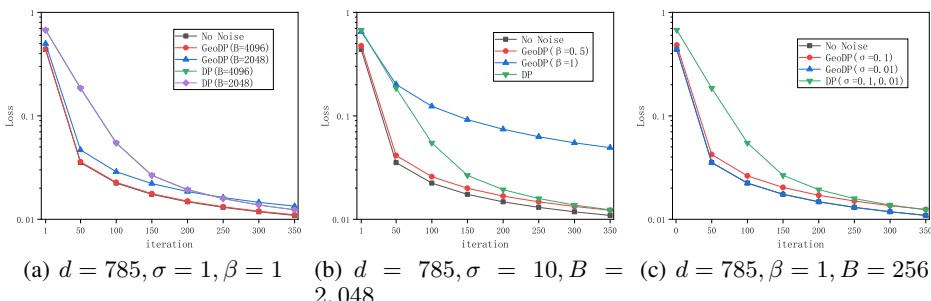

(a) $d = 785, \sigma = 1, \beta = 1$  (b) $d = 785, \sigma = 10, B = 2,048$  (c) $d = 785, \beta = 1, B = 256$

Figure 1: GeoDP versus DP on Logistic Regression under MNIST dataset

## 5 EXPERIMENTAL RESULTS

### 5.1 EXPERIMENTAL SETUP

We conduct our experiments on a server with Intel Xeon Silver 4210R CPU, 128G RAM, and Nvidia GeForce RTX 3090 GPU on Ubuntu 20.04 LTS system. All results are repeated 100 times to obtain the average. Unless otherwise specified, we fix $C = 0.1$. All codes are available in https://github.com/Derek0205/GeoDP.

For model efficiency, we use two prevalent benchmark datasets, **MNIST** (LeCun et al., 1998) and **CIFAR-10** (Krizhevsky et al., 2009). Besides, we also conduct a standalone experiment in Appendix D.2 to verify that GeoDP preserves directional information better than DP (Lemma 1). Due to the lack of public gradient datasets, we form a **synthetic dataset** for this experiment. To synthesize a dataset of gradients, we randomly collect $450,000$ gradients (of $20,000$ dimensions) from 9 epochs of training a non-DP CNN ($B = 1$) on CIFAR-10 (i.e., $50,000$ training images). Dimensions are randomly chosen in various experiments. Detailed information on datasets is in Appendix D.1.

As for models, we believe prevalent models such as **Logistic Regression (LR)**, **2-layer CNN** with Softmax activation and **ResNet** with 3 residual blocks (each one containing 2 convolutional layers and 1 rectified linear unit (ReLU)) are quite adequate to confirm the effectiveness of our strategy.

As for comparison methods, we compare GeoDP with DP on regular SGD from various perspectives, i.e., model efficiency, compatibility with existing optimization techniques. To demonstrate generality of GeoDP, we also apply a state-of-the-art clipping technique AUTO-S (Bu et al., 2024) to observe its improvements on GeoDP.

### 5.2 OVERALL EVALUATION

#### 5.2.1 GEODP VS. DP: LOGISTIC REGRESSION

The following experiments verify the effectiveness of GeoDP on Logistic Regression (LR) under MNIST dataset. Figure 1 plots training losses of 350 iterations, under *No noise*, *GeoDP* and *DP*. In Figure 1(a), with $B = 4,096$, GeoDP (the red line) significantly outperforms DP (the green line) and almost has the same performance as noise-free training (black line). The green line overlaps with the purple line because losses of DP-SGD with $B = 2,048$ and $B = 4,096$ are almost the same. This observation coincides with that from Figure 3(g), i.e., the batch size of DP-SGD hardly impacts the noise on the descent trend and thus the model efficiency. In contrast, batch size can successfully reduce the noise of GeoDP (see the gap between the red and blue lines).

In Figure 1(b), we test the performance of GeoDP under large noise scale. Initially, GeoDP (blue line) performs worse than DP (green line) with $\beta = 1$. When reducing $\beta$ to $0.5$ as suggested in Section D.2, the performance of GeoDP surges and leaves DP behind. This observation confirms the superiority of GeoDP over DP even under extreme cases.

In Figure 1(c), we fix the $\beta = 1$ and $B = 256$ while varying the noise multiplier in $\sigma = \{0.01, 0.1\}$. As we can see, reducing $\sigma$ cannot help DP to perform better (see the green line). This is because DP

introduces biased noise to the direction, as confirmed by Lemma 1. Simply reducing the variance of noise cannot counteract this bias. As such, **DP is sub-optimal even under very small multiplier.** By contrast, GeoDP can achieve significant efficiency improvement with multiplier reduction. When $\sigma = 0.01$ (see the blue line), GeoDP almost achieves noise-free model efficiency (the blue line is only slightly above the black line).

| Dataset | Method | $\sigma = 10$ | $\sigma = 1$ |
|---|---|---|---|
| MNIST (noise-free 99.11%) | DP ($B = 8192$) | 87.93% | 94.25% |
| | DP ($B = 16384$) | 88.12% | 95.52% |
| | DP($B = 16384$ +AUTO-S) | 88.40% | 95.71% |
| | GeoDP ($B = 8192, \beta = 0.1$) | 90.31% | 96.47% |
| | GeoDP ($B = 16384, \beta = 0.1$) | 93.58% | 98.04% |
| | GeoDP ($B = 8192, \beta = 0.5$) | 53.80% | 60.31% |
| | GeoDP ($B = 16384, \beta = 0.1$ +AUTO-S) | 93.64% | 98.17% |

Table 2: GeoDP vs. DP on CNN under MNIST Dataset: Test Accuracy

### 5.2.2 GEODP VS. DP: DEEP LEARNING

To demonstrate the effectiveness of GeoDP in various learning tasks, we also conduct experiments on MNIST dataset with Convolutional Neural Network (CNN) and Residual Networks (ResNet). Due to the extremely large number of parameters, we set the number of training epochs to 20. As for clipping, the noisy magnitude in GeoDP impacts the overall model efficiency, and therefore GeoDP also clips the magnitude before adding noise to it (see Step 6 in Algorithm 1). Since existing works (Bu et al., 2024; Zhang et al., 2022) clip the $L_2$-norm of the gradient (i.e., the magnitude), the same techniques can be applied to GeoDP. As such, GeoDP is general and can be integrated to the state-of-the-art clipping technique AUTO-S (Bu et al., 2024).

The main results are demonstrated in Table 2. GeoDP consistently outperforms DP under various parameters except for large $\beta$. We can observe that the test accuracy is dramatically decreased (e.g., $98.7\% \rightarrow 60.3\%$) when $\beta$ increases from 0.1 to 0.5. The reason behind is the extremely large sensitivity of GeoDP incurred by high dimensionality ($21,840$ dimensions), as discussed in D.2. Overall, we can always find such a $\beta$ ($\beta = 0.1$ in this experiment) that GeoDP outperforms DP in any task. Similar results in Table 3 of Appendix D.3 also demonstrates the effectiveness of GeoDP on ResNet under CIFAR-10 dataset.

## 6 LITERATURE REVIEW

As a privacy-preserving technique for training various models, DP-SGD is an adaptation of the traditional SGD algorithm to incorporate differentially private guarantees. Chaudhuri et al. 2011 initially introduced a DP-SGD algorithm for empirical risk minimization. Abadi et al. 2016 were one of the first to introduce DP-SGD into deep learning. Afterwards, DP-SGD has been rapidly applied to various models, such as generative adversarial network (Ho et al., 2021), Bayesian learning (Heikkilä et al., 2017), federated learning (Zhang et al., 2022), graph neural networks (Zhang et al., 2024b). More comprehensive related works on DP, SGD and DP-SGD are described in Appendix E.

## 7 CONCLUSION

In this work, we first theoretically analyze the impact of DP noise on the training process of SGD, which shows that the perturbation of DP-SGD is actually sub-optimal because it introduces biased noise to the direction. This inspires us to reduce the noise on direction for model efficiency improvement. We then propose our geometric perturbation mechanism GeoDP. Its effectiveness and generality are mutually confirmed by both rigorous proofs and experimental results. As for future work, we plan to study the impact of mainstream training optimizations, such as Adam optimizer (Tang et al., 2024), on GeoDP.

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
