APPENDIX

# A  ALGORITHM OF DP-SGD

---

**Algorithm 2** DP-SGD

---

**Require:** Batch size $B$, noise multiplier $\sigma$, clipping threshold $C$, learning rate $\eta$, total number of iterations $T$.
**Ensure:** Trained model parameters $\boldsymbol{w}_T$.
 1: Initialize a model with parameters $\boldsymbol{w}_0$.
 2: **for** each iteration $t = 0, 1, ..., T-2, T-1$ **do**
 3:    Derive the average clipped gradient $\tilde{\boldsymbol{g}}_t$ with respect to the sampled subset $S \in D$ and the clipping threshold $C$.
 4:    Add noise $\boldsymbol{n}_t$ drawn from a zero-mean Gaussian distribution with standard deviation $\sigma C \boldsymbol{I}$ to $\tilde{\boldsymbol{g}}_t$, i.e., $\boldsymbol{g}_t^* = \tilde{\boldsymbol{g}}_t + \boldsymbol{n}_t / B$, where $\boldsymbol{n}_t$ is jointly determined by both $\sigma$ and $C$.
 5:    Update $\boldsymbol{w}_{t+1}^*$ by taking a step in the direction of the noisy gradient, i.e., $\boldsymbol{w}_{t+1}^* = \boldsymbol{w}_t - \eta \boldsymbol{g}_t^*$.
 6: **end for**

---

# B  AN ILLUSTRATION OF HYPER-SPHERICAL COORDINATE SYSTEM

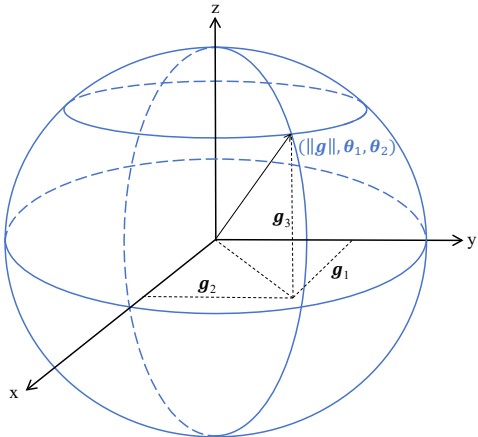

Figure 2: Coordinates Conversions in Three-dimensional Space.

# C  DETAILS OF PROOFS

## C.1  PROOF OF THEOREM .1

For DP-SGD, we have:

$$
\begin{aligned}
\left\| \boldsymbol{w}_{t+1}^* - \boldsymbol{w}^\star \right\|^2 &= \left\| \boldsymbol{w}_t - \boldsymbol{w}^\star - \eta \tilde{\boldsymbol{g}}_t^* \right\|^2 \\
&= \left\| \boldsymbol{w}_t - \boldsymbol{w}^\star \right\|^2 + \eta^2 \|\tilde{\boldsymbol{g}}_t^*\|^2 + 2\eta \langle \tilde{\boldsymbol{g}}_t^*, \boldsymbol{w}^\star - \boldsymbol{w}_t \rangle.
\end{aligned}
\tag{9}
$$

While for SGD, we have:

$$
\begin{aligned}
\left\| \boldsymbol{w}_{t+1} - \boldsymbol{w}^\star \right\|^2 &= \left\| \boldsymbol{w}_t - \boldsymbol{w}^\star - \eta \tilde{\boldsymbol{g}}_t \right\|^2 \\
&= \left\| \boldsymbol{w}_t - \boldsymbol{w}^\star \right\|^2 + \eta^2 \|\tilde{\boldsymbol{g}}_t\|^2 + 2\eta \langle \tilde{\boldsymbol{g}}_t, \boldsymbol{w}^\star - \boldsymbol{w}_t \rangle.
\end{aligned}
\tag{10}
$$

Subtracting Equation 10 from Equation 9, we have:

$$
\begin{aligned}
&\left\| \boldsymbol{w}_{t+1}^* - \boldsymbol{w}^\star \right\|^2 - \left\| \boldsymbol{w}_{t+1} - \boldsymbol{w}^\star \right\|^2 \\
=& \eta^2 \underbrace{\left( \|\tilde{\boldsymbol{g}}_t^*\|^2 - \|\tilde{\boldsymbol{g}}_t\|^2 \right)}_{Item\ \ A} + 2\eta \underbrace{\langle \tilde{\boldsymbol{g}}_t^* - \tilde{\boldsymbol{g}}_t, \boldsymbol{w}^\star - \boldsymbol{w}_t \rangle}_{Item\ \ B}.
\end{aligned}
\tag{11}
$$

Recall that $\boldsymbol{n}_t$ follows a noise distribution whose standard deviation is $C\sigma\boldsymbol{I}$. Suppose $\boldsymbol{n}_\sigma$ follows a noise distribution with the standard deviation $\sigma\boldsymbol{I}$, we have $\boldsymbol{n}_t = C\boldsymbol{n}_\sigma$. For Item A:

$$
\begin{aligned}
\|\tilde{\boldsymbol{g}}_t^*\|^2 - \|\tilde{\boldsymbol{g}}_t\|^2 &= (\tilde{\boldsymbol{g}}_t^* - \tilde{\boldsymbol{g}}_t)(\tilde{\boldsymbol{g}}_t^* + \tilde{\boldsymbol{g}}_t) \\
&= \boldsymbol{n}_t/B\,(2\tilde{\boldsymbol{g}}_t + \boldsymbol{n}_t/B) \\
&= 2\langle C\boldsymbol{n}_\sigma/B, \tilde{\boldsymbol{g}}_t\rangle + C^2\boldsymbol{n}_\sigma^2/B^2.
\end{aligned}
\tag{12}
$$

And for Item B:

$$
\tilde{\boldsymbol{g}}_t^* - \tilde{\boldsymbol{g}}_t = \boldsymbol{n}_t/B = C\boldsymbol{n}_\sigma/B.
\tag{13}
$$

Applying Equation 12 and 13 into Equation 11, we have:

$$
\begin{aligned}
&\left\|\boldsymbol{w}_{t+1}^* - \boldsymbol{w}^\star\right\|^2 - \left\|\boldsymbol{w}_{t+1} - \boldsymbol{w}^\star\right\|^2 \\
=&\eta^2\underbrace{\left(2\langle C\boldsymbol{n}_\sigma/B, \tilde{\boldsymbol{g}}_t\rangle + C^2\boldsymbol{n}_\sigma^2/B^2\right)}_{Item\quad A} + 2\eta C/B\underbrace{\langle\boldsymbol{n}_\sigma, \boldsymbol{w}^\star - \boldsymbol{w}_t\rangle}_{Item\quad B}.
\end{aligned}
\tag{14}
$$

## C.2 PROOF OF COROLLARY 1

Let us just assume DP-SGD reaches the global optima, i.e. $\boldsymbol{w}_t = \boldsymbol{w}^\star$. Accordingly, Item B becomes zero while Item A is non-zero unless $\boldsymbol{n}_\sigma$ stays zero (which is unlikely), as shown in Equation 15. That is, DP noise would immediately cause SGD to deviate from global optima even if SGD can reach optima.

$$
\lim_{\boldsymbol{w}_t\to\boldsymbol{w}^\star}\left\|\boldsymbol{w}_{t+1}^* - \boldsymbol{w}^\star\right\|^2 - \left\|\boldsymbol{w}_{t+1} - \boldsymbol{w}^\star\right\|^2 = \eta^2\underbrace{\left(\frac{2C}{B}\langle\boldsymbol{n}_\sigma, \tilde{\boldsymbol{g}}_t\rangle + \frac{C^2\boldsymbol{n}_\sigma^2}{B^2}\right)}_{Item\quad A}.
\tag{15}
$$

## C.3 PROOF OF COROLLARY 2

We analyze the effectiveness of DP-SGD techniques (i.e., fine-tuning clipping, learning rate and batch size) on Item A and Item B, respectively.

1. *Item A.*
   *As per learning rate,* we apply different learning rate $\eta^*$ to DP-SGD, and see if tuning $\eta^*$ can make Item A zero. Applying $\eta^*$ to Equation 11, we have:

   $$
   \text{Item A} = \eta^{*2}\|\tilde{\boldsymbol{g}}_t^*\|^2 - \eta^2\|\tilde{\boldsymbol{g}}_t\|^2.
   \tag{16}
   $$

   As Equation 16 is only composed of numerical values, fined-tuned $\eta^* = \eta^2\|\tilde{\boldsymbol{g}}_t\|^2/\|\tilde{\boldsymbol{g}}_t^*\|^2$ can certainly zero Item A.

   *As for clipping,* given $\boldsymbol{n}_\sigma$ is a random variable drawn from the noise distribution whose standard deviation is $\sigma\boldsymbol{I}$, we have:

   $$
   \boldsymbol{n}_t = C\boldsymbol{n}_\sigma.
   \tag{17}
   $$

   As $\tilde{\boldsymbol{g}}_t^* = \tilde{\boldsymbol{g}}_t + \boldsymbol{n}_t/B$, reducing $C$ certainly reduces the scale of $\tilde{\boldsymbol{g}}_t^*$. Overall, fine-tuning of DP-SGD can certainly reduce Item A.

2. *Item B.*
   *For learning rate,* we have:

   $$
   \begin{aligned}
   \text{Item B} &= \langle\eta^*\tilde{\boldsymbol{g}}_t^* - \eta\tilde{\boldsymbol{g}}_t, \boldsymbol{w}^\star - \boldsymbol{w}_t\rangle \\
   &= \|\eta^*\tilde{\boldsymbol{g}}_t^* - \eta\tilde{\boldsymbol{g}}_t\|\|\boldsymbol{w}^\star - \boldsymbol{w}_t\|\cos\theta.
   \end{aligned}
   \tag{18}
   $$

   where $\theta$ is the relative angle between two vectors. Apparently, no matter how to fine-tune $\eta^*$, how $\eta^*\tilde{\boldsymbol{g}}_t^* - \eta\tilde{\boldsymbol{g}}_t$ varies is rather random because there is no relevance between $\eta^*$ and $\eta^*\tilde{\boldsymbol{g}}_t^* - \eta\tilde{\boldsymbol{g}}_t$ as well as $\theta$.

   For clipping, we prove that it cannot change the geometric property of the perturbed gradient, although the noise scale is indeed changed. If the clipping thresholds $C_1$, $C_2$ and a gradient $\boldsymbol{g}(\|\boldsymbol{g}\| \geq C_1 \geq C_2)$, we have the clipped gradient $\tilde{\boldsymbol{g}}_1 = \frac{\boldsymbol{g}}{\|\boldsymbol{g}\|/C_1}, \tilde{\boldsymbol{g}}_2 = \frac{\boldsymbol{g}}{\|\boldsymbol{g}\|/C_2}$

as per Equation 4 and corresponding noise $\boldsymbol{n}_1 = C_1 \boldsymbol{n}_\sigma$, $\boldsymbol{n}_2 = C_2 \boldsymbol{n}_\sigma$ as per Equation 17. Accordingly, the perturbed gradient is:

$$\tilde{\boldsymbol{g}}_1^* = \tilde{\boldsymbol{g}}_1 + \boldsymbol{n}_1/B = \frac{\boldsymbol{g}}{\|\boldsymbol{g}_1\|/C_1} + C_1/B\boldsymbol{n}_\sigma.$$

$$\tilde{\boldsymbol{g}}_2^* = \tilde{\boldsymbol{g}}_2 + \boldsymbol{n}_2/B = \frac{\boldsymbol{g}}{\|\boldsymbol{g}_2\|/C_2} + C_2/B\boldsymbol{n}_\sigma. \tag{19}$$

Then, we have:

$$\frac{\tilde{\boldsymbol{g}}_1^*}{C_1} = \frac{\tilde{\boldsymbol{g}}_2^*}{C_2}.$$

$$\|\tilde{\boldsymbol{g}}_1^*\| \geq \|\tilde{\boldsymbol{g}}_2^*\|. \tag{20}$$

Namely, clipping cannot control the directions of perturbed gradients $\frac{\tilde{\boldsymbol{g}}_1^*}{C_1} = \frac{\tilde{\boldsymbol{g}}_2^*}{C_2}$, while indeed reducing the noise scale ($\|\tilde{\boldsymbol{g}}_1^*\| \geq \|\tilde{\boldsymbol{g}}_2^*\|$).

### C.4 PROOF OF THEOREM .2

$\{\tilde{\boldsymbol{g}}_j | 1 \leq j \leq B\}$ are independently and identically distributed variables because each one is derived from one data $s_j$ of the same subset $S$. According to *CLT*, the following probability holds:

$$\lim_{B \to \infty} \Pr\left(\frac{\sum_{j=1}^B \tilde{\boldsymbol{g}}_{jz} - B * \mathbb{E}(\tilde{\boldsymbol{g}}_{jz})}{\sqrt{B * var(\tilde{\boldsymbol{g}}_{jz})}} \leq X\right)$$

$$= \lim_{B \to \infty} \Pr\left(\frac{\frac{1}{B}\sum_{j=1}^B \tilde{\boldsymbol{g}}_{jz} - \mathbb{E}(\tilde{\boldsymbol{g}}_{jz})}{\sqrt{var(\tilde{\boldsymbol{g}}_{jz})/B}} \leq X\right) = \int_{-\infty}^X \phi(x)dx, \tag{21}$$

where $\phi(x) = \frac{1}{\sqrt{2\pi}}\exp(-\frac{x^2}{2})$ is the pdf of the standard Gaussian distribution. As such, $\frac{\sum_{j=1}^B \tilde{\boldsymbol{g}}_{jz}/B - \mathbb{E}(\tilde{\boldsymbol{g}}_{jz})}{\sqrt{var(\tilde{\boldsymbol{g}}_{jz})/B}}$ follows standard Gaussian distribution $\mathcal{N}(0, 1)$, by which this theorem is proved.

### C.5 PROOF OF LEMMA 1

For traditional DP (adding noise $\boldsymbol{n}$ to the gradient $\boldsymbol{g}$), we can derive the perturbed angle $\boldsymbol{\theta}_z^*$ according to Equation 6, i.e.,

$$\boldsymbol{\theta}_z^* = \begin{cases} \arctan2\left(\sqrt{\sum_z^{d-1}(\boldsymbol{g}_{z+1} + \boldsymbol{n}_{z+1})^2}, \boldsymbol{g}_z + \boldsymbol{n}_z\right) & \text{if } 1 \leq z \leq d-2, \\ \arctan2\left(\boldsymbol{g}_{z+1} + \boldsymbol{n}_{z+1}, \boldsymbol{g}_z + \boldsymbol{n}_z\right) & \text{if } z = d-1. \end{cases} \tag{22}$$

Observing both acrtan2 equations above, we can conclude that the **traditional DP perturbation** introduces **biased** noise to the original direction, i.e., $\mathbb{E}(\boldsymbol{\theta}^*) \neq \boldsymbol{\theta}(bias(\boldsymbol{\theta}^*) \neq 0)$. Also, the variance of $\boldsymbol{\theta}$ ($var(\boldsymbol{\theta}^*)$) is non-zero, if the noise scale $\boldsymbol{n}_\sigma > 0$.

For GeoDP, we have $\boldsymbol{\theta}^\star = \boldsymbol{\theta} + \frac{\sqrt{d+2}\beta\pi}{B}\boldsymbol{n}_\sigma$. Accordingly, $\mathbb{E}(\boldsymbol{\theta}^\star) = \mathbb{E}(\boldsymbol{\theta} + \frac{\sqrt{d+2}\beta\pi}{B}\boldsymbol{n}_\sigma) = \boldsymbol{\theta}(bias(\boldsymbol{\theta}^\star) = 0)$, which means that GeoDP adds unbiased noise to the direction. Besides, $beta$ directly controls the noise added to the direction. In specific, the variance of $\boldsymbol{\theta}^\star(var(\boldsymbol{\theta}^\star))$ can approaching zero if $\beta \to 0$, because $\boldsymbol{\theta}^\star = \boldsymbol{\theta} + \frac{\sqrt{d+2}\beta\pi}{B}\boldsymbol{n}_\sigma$ approaches 0 if $\beta \to 0$.

Given that $\text{MSE}(\boldsymbol{\theta}) = bias^2(\boldsymbol{\theta}) + var(\boldsymbol{\theta})$ (Duan et al., 2024), there always exist such one $\beta$ that:

$$\text{MSE}(\boldsymbol{\theta}^\star) = bias^2(\boldsymbol{\theta}^\star) + var(\boldsymbol{\theta}^\star) <= bias^2(\boldsymbol{\theta}^*) + var(\boldsymbol{\theta}^*) = \text{MSE}(\boldsymbol{\theta}^*). \tag{23}$$

by which this lemma is proven.

### C.6 PROOF OF THEOREM .3

Following Corollary 2, we just have to prove Item B of GeoDP is smaller than Item A of DP. Different learning rates $\eta^\star$ and $\eta^*$ are applied to GeoDP and DP, respectively. Recall from Corollary

2, we have:

$$\begin{aligned}
\text{Item B} &= \langle \eta^\star \tilde{\boldsymbol{g}}_t^\star - \eta \tilde{\boldsymbol{g}}_t, \boldsymbol{w}^\star - \boldsymbol{w}_t \rangle \\
&= \underbrace{\|\eta^\star \tilde{\boldsymbol{g}}_t^\star - \eta \tilde{\boldsymbol{g}}_t\|}_{C} \underbrace{\|\boldsymbol{w}^\star - \boldsymbol{w}_t\|}_{D} \underbrace{\cos\theta}_{E}.
\end{aligned} \tag{24}$$

Note that the only way to optimize Item B is via Item $C$, whereas Item $D$, the distance between the current model and the optima (this distance is a vector), is fixed, and Item E, the relative angle between noise and the fixed distance, is too random. Therefore, we should reduce Item C as much as possible to optimize Item B. In general, we have:

$$\text{Item C}^2 = (\eta^\star \tilde{\boldsymbol{g}}_t^\star)^2 + (\eta \tilde{\boldsymbol{g}}_t)^2 - 2\eta^\star \eta \langle \tilde{\boldsymbol{g}}_t^\star, \tilde{\boldsymbol{g}}_t \rangle. \tag{25}$$

While $(\eta^\star \tilde{\boldsymbol{g}}_t^\star)^2 + (\eta \tilde{\boldsymbol{g}}_t)^2$ can be fine-tuned to zero by the learning rates, the only way for $\langle \tilde{\boldsymbol{g}}_t^\star, \tilde{\boldsymbol{g}}_t \rangle$ to be zero is that the direction of $\boldsymbol{g}^\star$ approximates that of $\tilde{\boldsymbol{g}}_t$ (or the opposite direction of $\tilde{\boldsymbol{g}}_t$, which rarely happens and is therefore ignored here). Since $\text{MSE}(\hat{\boldsymbol{\theta}}_t^\star) < \text{MSE}(\tilde{\boldsymbol{\theta}}_t^*)$ in Lemma 1, GeoDP therefore makes Item B zero more easily than DP, by which our theorem is proved.

# D SUPPLEMENTARY INFORMATION ON EXPERIMENTS

This section provides extensive information on experiments.

## D.1 DATASETS

**MNIST.** This is a dataset of 70,000 gray-scale images (28x28 pixels) of handwritten digits from 0 to 9, commonly used for training and testing machine learning algorithms in image recognition tasks. It consists of 60,000 training images and 10,000 testing images, with an even distribution across the 10 digit classes.

**CIFAR-10.** It is a dataset of 60,000 small (32x32 pixels) color images, divided into 10 distinct classes such as animals and vehicles, used for machine learning and computer vision tasks. It contains 50,000 training images and 10,000 testing images, with each class having an equal number of images.

## D.2 GEODP VS. DP: ACCURACY OF DESCENT TREND

We verify the superiority of GeoDP on preserving directional information. On the synthetic dataset, we perturb gradients by GeoDP and DP, respectively, and compare their MSEs under various parameters. As illustrated in Figure 3, labels $\theta$ and $g$ represent MSEs of perturbed directions and gradients, respectively. In Figure 3(a)-3(c), we fix dimension $d = 5,000$ and batch size $B = 2,048$, while varying noise multiplier $\sigma$ in $\{10^{-4}, 10^{-3}, 10^{-2}, 10^{-1}, 1, 10\}$ if $\delta = 10^{-5}$) under three bounding factors $\beta = \{0.01, 0.1, 1\}$, respectively. We have two major observations. First, GeoDP better preserves directions (the red line is below the black line) while DP better preserves gradients (the blue line is below the green line) in most scenarios. Second, GeoDP is sometimes not robust to large noise multiplier and high dimensionality. When $\sigma > 1$ in Figure 3(a), GeoDP is instead outperformed by DP in preserving directions. Similar results can be also observed in Figure 3(d)-3(f) (fixing $\sigma = 8, B = 4096$ while varying dimensionality in $\{500, 1000, 2000, 5000, 10000, 20000\}$) and Figure 3(g)-3(i) (fixing $d = 10000, \sigma = 8$ while varying batch size in $\{512, 1024, 2048, 4096, 8192, 163984\}$), respectively. For example, Figure 3(d) and Figure 3(g), which all fix $\beta = 1$, show that GeoDP is outperformed by DP on preserving directions when $d > 2000$ and $B < 8192$, respectively.

Before addressing this problem, we discuss reasons behind the ineffectiveness of GeoDP. Recall from Section 4.2 that the perturbation of GeoDP on directions is $\frac{\sqrt{d+2}\beta\pi}{B}\boldsymbol{n}_\sigma$. Obviously, both large noise multiplier ($\boldsymbol{n}_\sigma$) and high dimensionality ($\sqrt{d+2}$) increase the perturbation on directions.

Nevertheless, GeoDP can overcome this shortcoming by tuning $\beta$, which controls the sensitivity of direction. In both Figures 3(b) ($\beta = 0.1$) and 3(c) ($\beta = 0.01$), we reduce the noise on the direction by reducing the bounding factor, and the pay-off is very significant. Results show that GeoDP simultaneously outperforms DP in both direction and gradient. Tuning $\beta$ is also effective

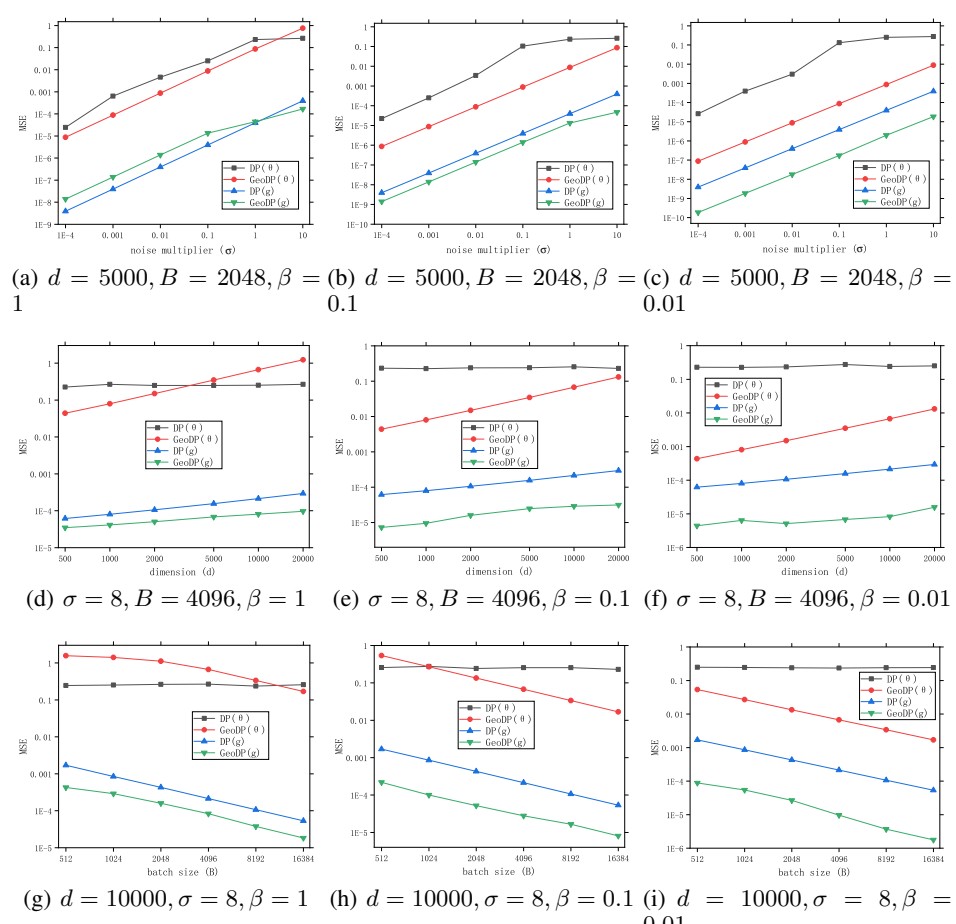

Figure 3: GeoDP vs. DP on Preserving Gradients under Various Parameters on Synthetic Dataset

in Figure 3(e), 3(f) and Figure 3(h), 3(i), respectively. Most likely, smaller bounding factor reduces noise added to the direction while does not affect the noisy magnitude. Accordingly, GeoDP reduces both MSEs of direction and gradient, and thus perfectly outperforms DP in preserving directional information.

To further confirm this conjecture, extensive experiments, by varying the bounding factor in $\{0.1, 0.2, 0.4, 0.6, 0.8, 1.0\}$ under different scenarios, are conducted in Figure 4. All experimental results show that there always exists a bounding factor ($\beta = 0.2$ in Figure 4(a) and $\beta = 0.4$ in Figure 4(b)) for GeoDP to outperform DP in preserving both direction and gradient. **These results also perfectly align with our theoretical analysis in Lemma 1 and Theorem .3, respectively.**

Also, GeoDP can improve accuracy by tuning batch size. As illustrated in Figure 3(g) ($d = 10000, \sigma = 8, \beta = 1$), we demonstrate how the performance of GeoDP is impacted by batch size. Obviously, a large batch size can boost GeoDP to provide optimal accuracy on directions. In contrast, the accuracy of DP on directions hardly changes with batch size (see the black line in 3(g)), although the noise scale on gradients is reduced by larger batch size (see the blue line in 3(g)). These results validate that **optimization techniques of DP-SGD**, such as fine-tuning learning rate, clipping threshold and batch size, **cannot reduce the noise on the direction, as confirmed by Corollary 2.**

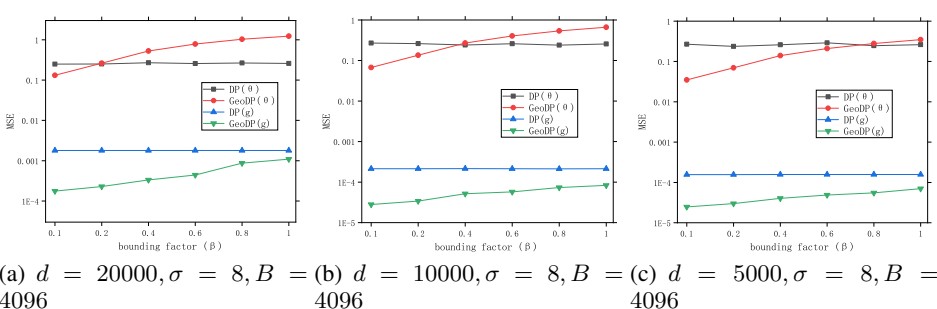

(a) $d = 20000, \sigma = 8, B = 4096$    (b) $d = 10000, \sigma = 8, B = 4096$    (c) $d = 5000, \sigma = 8, B = 4096$

Figure 4: The Effectiveness of Bounding Factor

### D.3 GeoDP vs. DP: ResNet

Similar to our observations on LR, GeoDP even outperforms DP under smaller noise multiplier (e.g., under $\beta = 1$.). Note that the perturbed direction of GeoDP is unbiased while that of DP is biased, as previously confirmed in Lemma 1. As such, the optimality of GeoDP over DP under smaller noise is a reflection of this property.

| Dataset | Method | $\sigma = 0.1$ | $\sigma = 0.01$ |
|---|---|---|---|
| | DP ($B = 8192$) | 59.39% | 63.27% |
| | DP ($B = 16384$) | 60.12% | 63.84% |
| | DP($B = 16384$ +AUTO-S) | 60.51% | 63.91% |
| CIFAR-10 (noise-free 67.43%) | GeoDP ($B = 8192, \beta = 1$) | 61.47% | 65.93% |
| | GeoDP ($B = 16384, \beta = 1$) | 63.38% | 66.51% |
| | GeoDP ($B = 16384, \beta = 0.1$) | 65.47% | 67.35% |
| | GeoDP ($B = 16384, \beta = 0.1$ +AUTO-S) | 65.58% | 67.37% |

Table 3: GeoDP vs. DP on ResNet under CIFAR-10 Dataset: Test Accuracy

## E  Literature review

In this section, we review related works from three aspects: DP, SGD and their crossover works DP-SGD.

### E.1  Differential Privacy (DP)

DP (Dwork et al., 2014; Wasserman & Zhou, 2010) is a framework designed to provide strong privacy guarantees for datasets whose data is used in data analysis or machine learning models. It aims to allow any third party, e.g., data scientists and researchers, to glean useful insights from datasets while ensuring that the privacy of individuals cannot be compromised. Since Dwork *et al.* 2006 first introduced the definition of *differential privacy* (DP), DP has been extended to various scopes, such as numerical data collection (Duchi et al., 2018; Wang et al., 2019), set-value data collection (Chen et al., 2011; Wang et al., 2020), key-value data collection (Ye et al., 2021b), high-dimensional data (Duan et al., 2022), graph analysis (Sun et al., 2023), time series data release (Ye et al., 2021a), private learning (Zheng et al., 2019; Fu et al., 2023), federated matrix factorization (Li et al., 2021), data mining (Hu et al., 2015), local differential privacy (Xu et al., 2020; 2019; Bao et al., 2021; Wang et al., 2018), database query (Farias et al., 2023; Bogatov et al., 2021), Markov model (Xiao et al., 2017) and benchmark (Schäler et al., 2023; Duan et al., 2022; 2024). Relevant to our work, we follow the common practice to implement Gaussian mechanism (Dwork et al., 2014) to perturb model parameters. Besides, Rényi Differential Privacy (RDP) (Mironov, 2017) allows us to more accurately estimate the cumulative privacy loss of the whole training process.

### E.2 STOCHASTIC GRADIENT DESCENT (SGD)

Stochastic Gradient Descent (SGD) is a fundamental optimization algorithm widely used in machine learning and deep learning for training a wide array of models. It is especially popular for its efficiency in dealing with large datasets and high-dimensional optimization problems. SGD was first introduced by Herbert *et al.* 1951, and applied for training deep learning models 1986. The development of SGD has seen several significant improvements over the years. Xavier *et al.* 2010 and Yoshua 2012 optimized deep neural networks using SGD. Momentum, a critical concept to accelerate SGD, was emphasized by Llya *et al.* 2013. Diederik *et al.* 2015 proposed Adam, a variant of SGD that adaptively adjusts the learning rate for each parameter. Sergey *et al.* 2015 introduced Batch Normalization, a technique to reduce the internal covariate shift in deep networks. Yang *et al.* 2017 and Zhang *et al.* 2019 further proposed large-batch training and lookahead optimizer, respectively. These advancements have pushed the boundaries of SGD, enabling efficient training of increasingly complex deep learning models (Xu et al., 2024; Zhang et al., 2024a; Wang et al., 2024; Xing et al., 2024). Without loss of generality, we follow the common practice of existing works and implement SGD without momentum to better demonstrate the efficiency of our strategy.

### E.3 DIFFERENTIALLY PRIVATE STOCHASTIC GRADIENT DESCENT (DP-SGD)

As a privacy-preserving technique for training various models, DP-SGD is an adaptation of the traditional SGD algorithm to incorporate differentially private guarantees. This is crucial in applications where data confidentiality and user privacy are concerns, such as in medical or financial data processing. The basic idea is adding DP noise to gradients during the training process. Chaudhuri et al. 2011 initially introduced a DP-SGD algorithm for empirical risk minimization. Abadi et al. 2016 were one of the first to introduce DP-SGD into deep learning. Afterwards, DP-SGD has been rapidly applied to various models, such as generative adversarial network (Ho et al., 2021), Bayesian learning (Heikkilä et al., 2017), federated learning (Zhang et al., 2022), graph neural networks (Zhang et al., 2024b).

As for optimizing model efficiency of DP-SGD, there are three major streams. First, gradient clipping can help to reduce the noise scale while still following DP framework. For example, adaptive gradient clipping (Xia et al., 2023; Zhang et al., 2022; Chen et al., 2020), which adaptively bounds the sensitivity of the DP noise, can trade the clipped information for noise reduction. Second, we can amplify the privacy bounds to save privacy budgets, such as Rényi Differential Privacy (Gopi et al., 2021). Last, more efficient SGD algorithms, such as DP-Adam (Tang et al., 2024), can be introduced to DP-SGD so as to improve the training efficiency.

However, existing works still cling to numerical perturbation, and there is no work investigating whether the numerical DP scheme is optimal for the geometric SGD in various applications. In this work, we instead fill in this gap **by proposing a new DP perturbation scheme**, which exclusively preserves directions of gradients so as to improve model efficiency. As no previous works carry out optimization from this perspective, **our work is therefore only parallel to vanilla DP-SGD while orthogonal to all existing works**.