# OpenReview forum: "Analyzing and Optimizing Perturbation of DP-SGD Geometrically"
_ICLR.cc/2025/Conference — ICLR 2025 Conference Withdrawn Submission_

### Official Review · Reviewer_iMf4 · 2024-10-23

**Soundness:** 2
**Presentation:** 2
**Contribution:** 2
**Rating:** 5
**Confidence:** 4

**Summary:**

This paper proposes GeoDP, a modification of the DP-SGD algorithm, aiming to address direction bias introduced by noise. The core idea is to transform the gradient into hyper-spherical coordinates, apply Differential Privacy (DP) in this transformed space, and then revert back to update the model. The DP mechanism follows a similar process to DP-SGD, involving bounding (clipping) and noise addition.

**Strengths:**

1. The paper is well-motivated in addressing the directional bias
2. The paper is easy to follow, and the code is publicly available

**Weaknesses:**

1. **Unclear Novelty**: The novelty of GeoDP remains unclear. While the authors claim their method remedies the direction bias of gradients, they still adopt gradient clipping, which inherently introduces bias. Furthermore, the gradient direction is still bounded by the parameter $\beta$.

    To clarify the novelty, it would be helpful if the authors could provide a more detailed comparison of the bias introduced by GeoDP’s bounding versus traditional gradient clipping. Specifically, how does the noise introduced by GeoDP differ in terms of bias compared to DP-SGD, given that both methods use some form of bounding? A deeper analysis of this would help demonstrate how GeoDP introduces "unbiased" noise or mitigates bias more effectively.


2. **Weak Evaluation**: The evaluation lacks depth, particularly due to the omission of the privacy parameter $\epsilon$, which hinders comparison with other DP methods in the literature. Including the value of $\epsilon$ in the evaluation would allow for a clearer comparison of privacy guarantees across methods.

    Additionally, since GeoDP introduces the new hyperparameter $\beta$, it would be beneficial to provide a sensitivity analysis of both $\beta$ and the gradient clipping parameter $C$. This analysis should illustrate how tuning each parameter impacts model performance. Such a comparison would clarify which parameter plays a more critical role in optimizing the algorithm's effectiveness.

**Questions:**

1. **Example 1**: The example provided does not adequately demonstrate the directional bias in DP-SGD. The cases being compared are:
    - Case 1: $g+n_1$ — no gradient clipping + noise
    - Case 2: $g/2 + n_1/2$ — gradient clipping + rescaled noise

    Since DP-SGD typically involves gradient clipping, a more relevant comparison would involve:

    - Case 1: $g/C_1 + n_1$ — gradient clipping + noise
    - Case 2: $g/C_2 + n_1/(C_1/C_2)$ — gradient clipping + rescaled noise

    This revised comparison would more effectively demonstrate the directional bias introduced by DP-SGD. I would suggest revising Example 1 to include these cases, as they provide a clearer illustration of how clipping affects gradient direction.


1. **Definition 4**: When referring to "m gradients," are you referring to gradients from "m iterations"? If so, measuring the mean squared error (MSE) of gradient directions over multiple iterations may not be the best approach, as gradient directions naturally vary during training, even with identical initialization, data, and hyperparameters.

    To provide a more meaningful evaluation, I suggest considering alternative metrics that account for this natural variation. For example, you could compare the angular difference between the average gradient direction and the true gradient direction over a fixed number of iterations. This would offer a more consistent measure of directional stability and bias across iterations.


1. **Example 2**: The description of noise accumulation in three dimensions $(n_1, n_2, n_3)$ and how this results in a biased first angle $\theta_1$ is unclear. Could you clarify how this bias manifests in the accumulation? You mention that $\mathbb{E}(\theta_1^*) \neq \theta_1$, but the reasoning behind this is not fully explained.

    To make this clearer, I suggest providing a mathematical derivation or empirical demonstration showing that $\mathbb{E}(\theta_1^*) \neq \theta_1$. This would offer a more rigorous and transparent illustration of how the bias in the first angle arises due to noise accumulation.


1. **Example 3**: Could you provide an example, similar to Example 3, demonstrating how GeoDP maintains the direction of the perturbed gradient?

    To make this comparison more insightful, I suggest adding a parallel example that shows how GeoDP perturbs the gradient and comparing the resulting directions side-by-side with traditional DP-SGD. This would give readers a clearer understanding of the advantages GeoDP offers in preserving gradient direction compared to DP-SGD.


1. **Related Work**: There's another paper that discusses the impact of DP on gradient direction: *Protection Against Reconstruction and Its Applications in Private Federated Learning* (https://arxiv.org/pdf/1812.00984). How does your work compare with theirs in terms of addressing the gradient direction issue when applying DP?

    To strengthen the related work section, I suggest providing a brief comparison between your approach and the one in this paper. Highlight the key similarities and differences, particularly in how each method tackles the gradient direction issue. This would give readers a better context for understanding the contributions of GeoDP relative to existing methods.


1. **Code**: In the `cartesian_to_hyperspherical` function in the CNN.py file, why did you use `torch.acos` and `torch.atan` instead of `torch.atan2`, as mentioned in Equation 6? Is this an equivalent operation, or is there a reason for this difference?

    To clarify this discrepancy, I suggest explaining the rationale behind using `torch.acos` and `torch.atan` instead of `torch.atan2`. Does this affect the algorithm’s performance or theoretical guarantees in any way? If there is no specific reason for this difference, it might be helpful to update the implementation to match the equation in the paper for consistency and transparency.

---

### Official Review · Reviewer_FTyK · 2024-11-02

**Soundness:** 1
**Presentation:** 3
**Contribution:** 1
**Rating:** 1
**Confidence:** 4

**Summary:**

This paper addresses limitations in differentially private stochastic gradient descent (DP-SGD) by optimizing it from a geometric perspective. DP-SGD is widely used to add privacy-preserving noise to gradients during training; however, this paper highlights that this process introduces biased noise in gradient directions, which in turn hampers model efficiency. The authors propose a novel method, GeoDP, which separately perturbs the gradient’s direction and magnitude. By using a hyper-spherical coordinate system, GeoDP represents gradient size and direction through radius and angles. The paper provides both theoretical analyses and empirical results on datasets such as MNIST and CIFAR-10, demonstrating that GeoDP substantially enhances efficiency and accuracy compared to standard DP-SGD.

**Strengths:**

# Novel Intuition:
The paper presents a novel intuition by approaching the optimization of DP-SGD through geometric perturbation. This idea of separating gradient direction and magnitude for noise addition is creative, and it brings a fresh perspective to improving privacy-preserving methods.

# Significance of the Problem:
The paper tackles a highly important issue in the domain of privacy-preserving deep learning. DP-SGD plays a crucial role in safeguarding data privacy in training, but its primary limitation has been the reduction in model utility due to noise. This paper's attempt to directly address the efficiency loss in DP-SGD is therefore extremely valuable.

# Clear and Logical Presentation:
The paper is logically structured and easy to follow. The authors present their motivation, methodology, and results in a straightforward manner, making it accessible even to readers who are less familiar with the mathematical complexities of DP and SGD. This clarity in exposition enhances the overall impact and readability of the work.

**Weaknesses:**

# Incorrect Implementation Affecting Understanding and Validity of Results:
After reviewing the authors' submitted code, I found that when training the CNN with PyTorch, the clipping operation was applied to the averaged gradient. However, in a correct DP-SGD implementation, each sample’s gradient should be computed first, then clipped individually before averaging. This process is crucial to DP-SGD, as it limits each sample’s contribution to the gradient, distinguishing DP-SGD from standard SGD. Previous studies [1] have shown that per-sample clipping has a substantial impact on optimization results, so this is a significant oversight. This incorrect implementation raises concerns about the authors' understanding of DP-SGD and renders the subsequent experimental results unreliable. I suggest that the authors use a correct implementation and re-run the experiments, potentially utilizing the Opacus package to compute per-sample gradients in pytorch.

# Unclear Determination of Direction Range:
The algorithm uses a “Direction range” (line 318), but how the values for \Gamma_1 and \Gamma_2 were chosen is not clearly explained in the paper. I also could not locate this implementation in the code (though I may have missed it). Without a proper range for controlling direction, the sensitivity calculation may be incorrect, implying that the noise added could be insufficient to meet DP requirements, leading to an uncorrectable privacy error. I recommend that the authors clarify the determination of \Gamma_1 and \Gamma_2 and ensure their implementation is reflected in the code.

# Limited Experimental Comparisons:
The experimental section lacks comparisons with recent DP research, failing to show results near SOTA performance. As DP-SGD is highly sensitive to hyperparameter choices, comparing the proposed method with recent research would lend more credibility than showing specific cases, such as \sigma = 10 or \sigma = 1, which may not be representative. I recommend that the authors benchmark their method against more recent SOTA DP approaches to robustly validate their improvements [2].

# Lack of Privacy Budget Reporting:
The paper does not report the privacy budget parameters \epsilon and \delta. These parameters are critical to understanding the strength of the privacy guarantees offered by the method. I suggest that the authors include the privacy budget in their experimental results to provide a more complete picture.


[1] Bu, Z., Wang, H., Dai, Z., & Long, Q. (2023). On the convergence and calibration of deep learning with differential privacy. Transactions on machine learning research, 2023.
[2] De, S., Berrada, L., Hayes, J., Smith, S. L., & Balle, B. (2022). Unlocking high-accuracy differentially private image classification through scale. arXiv preprint arXiv:2204.13650.

**Questions:**

Please refer to the issues raised above. I consider points 1-3 to be critical. If I have misunderstood anything, I would be happy to hear the authors' explanations.

---

### Official Review · Reviewer_X3rA · 2024-11-03

**Soundness:** 3
**Presentation:** 3
**Contribution:** 4
**Rating:** 6
**Confidence:** 3

**Summary:**

This paper has two main contributions:
  1. Theoretically proved that DP-SGD is sub-optimal in model efficiency
  2. Proposed Geo-DP-SGD, which improved the performance of DP-SGD without losing privacy.
Besides, the results have been verified with experiments on real-world datasets.

**Strengths:**

The theoretical analysis is beautiful and the proposed algorithm has been experimentally verified.

**Weaknesses:**

1. The setting of $\beta$ parameter is not well discussed.
2. The related works are not well discussed.
3. The presentation needs to be improved. It's not clear enough in Algorithm 1 how the dimensions are decided.

**Questions:**

Any analysis about how $\beta$ should be decided in real-world applications?

---

### Official Review · Reviewer_GxkM · 2024-11-04

**Soundness:** 1
**Presentation:** 1
**Contribution:** 2
**Rating:** 3
**Confidence:** 4

**Summary:**

The paper proposes to decompose the gradient noise in DP-SGD into separate noise parts for gradient magnitude and direction. The paper argues that this makes DP-SGD gradient updates unbiased, and improves performance over DP-SGD at the same level of noise on MNIST with logistic regression and convolutional nets.

**Strengths:**

This is a very interesting idea, and I am quite convinced that separating the noise into magnitude and direction components can significantly improve the convergence properties of DP-SGD.

**Weaknesses:**

Despite the idea being very promising, I think there are several critical issues in the current version of the manuscript:
- Most importantly, lack of privacy analysis. What are the DP guarantees of the algorithm? Do we have to count each gradient update as an application of two Gaussian mechanisms? If not, why? Without answers to this, we cannot compare to standard DP-SGD with the same level of noise as the privacy guarantees are likely different at the same noise scale. The paper promises a privacy analysis in Section 4.3, but it is not there.
- Insufficient evaluation:
  - MNIST is not a representative dataset, so we cannot really know how much better is GeoDP in many practical scenarios. I would suggest including at least CIFAR10 and, e.g., language model finetuning on MMLU.
  - I would also suggest to systematically compare with different clipping strategies.
- The proofs of formal results are not included. The appendix is missing in the paper, and there is no supplement.

Minor issues:
- The wording switches between efficiency and deficiency seemingly. It would help to have consistent naming.
- There are multiple typos. The manuscript would benefit from careful proofreading.
- Theorem 2 has a broken reference as "Theorem .2"

In sum, I think the idea is very promising, but the paper would need a formal privacy analysis, additional experimental evaluations, and fix multiple presentation issues to pass the bar.

**Questions:**

- What are the privacy guarantees of the algorithm?

---

### Official Review · Reviewer_5Fcm · 2024-11-05

**Soundness:** 1
**Presentation:** 1
**Contribution:** 1
**Rating:** 1
**Confidence:** 5

**Summary:**

The paper proposes a way to refine DP-SGD which first converts the average gradient to certain space, add noise, then convert back to Euclidean space.

**Strengths:**

The topic of improving DP-SGD is important and trending.

**Weaknesses:**

It seems to be an LLM-generated paper ...

The proposed algorithm's biggest problem is that there's no proof of the privacy guarantee. I don't see how Step 2 in the algorithm in Sec 4.2 restricts the sensitivity. The paper is poorly written. I took a look at Theorem 1 and I think it is completely useless. The proof seems to assume SGD also uses a clipped gradient, and both DP-SGD and SGD start with w_t. I suspect the entire theorem is generated by GPT.

**Questions:**

Is this an LLM-generated paper?

---

### Note · Authors · 2025-01-22

**Comment:**

Due to the low initial rating, we decide to withdraw this submission and skip the rebuttal phase. Thank you for your understanding.

**Withdrawal Confirmation:**

I have read and agree with the venue's withdrawal policy on behalf of myself and my co-authors.